# Dietary Patterns and Major Depression: Results from 15,262 Participants (International ALIMENTAL Study)

**DOI:** 10.3390/nu17091583

**Published:** 2025-05-04

**Authors:** Yannis Achour, Guillaume Lucas, Sylvain Iceta, Mohamed Boucekine, Masoud Rahmati, Michael Berk, Tasnime Akbaraly, Bruno Aouizerate, Lucile Capuron, Wolfgang Marx, Melissa M. Lane, Cao Duy Nguyen, Huyen Do, Bach Xuan Tran, Dong Keon Yon, Laurent Boyer, Guillaume Fond

**Affiliations:** 1APHM, CEReSS, Research Centre on Health Services and Quality of Life, Hôpitaux Universitaires de Marseille, Aix Marseille University, 13005 Marseille, France; 2APHM, Département Universitaire de Médecine Générale (DUMG), Aix Marseille University, 13005 Marseille, France; 3Research Center of the Quebec Heart and Lung Institute, Quebec City, QC G1V 4G5, Canada; 4Department of Psychiatry and Neurosciences, Laval University, Quebec City, QC G1V 0A6, Canada; 5Fondamental Fondation, 94000 Créteil, France; 6Department of Physical Education and Sport Sciences, Faculty of Literature and Human Sciences, Lorestan University, Khoramabad 6815144316, Iran; 7Department of Physical Education and Sport Sciences, Faculty of Literature and Humanities, Vali-E-Asr University of Rafsanjan, Rafsanjan 7718897111, Iran; 8School of Medicine, Barwon Health, Institute for Mental and Physical Health and Clinical Translation (IMPACT), Deakin University, Geelong, VIC 3216, Australia; 9Orygen, National Centre of Excellence in Youth Mental Health, University of Melbourne, Melbourne, VIC 3010, Australia; 10Florey Institute for Neuroscience and Mental Health, Department of Psychiatry, University of Melbourne, Melbourne, VIC 3084, Australia; 11Desbrest Institute of Epidemiology and Public Health (IDESP), University of Montpellier, INSERM (Institut National de Santé et de la Recherche Médicale), 34090 Montpellier, France; 12Regional Reference Center for the Management and Treatment of Anxiety and Depressive Disorders, Charles Perrens Hospital, 33076 Bordeaux, France; 13INRAE, Bordeaux INP, NutriNeuro, University of Bordeaux, UMR 1286, 33000 Bordeaux, France; 14Institute for Global Health Innovations, Duy Tan University, Da Nang 550000, Vietnam; nguyencaoduy@duytan.edu.vn; 15Research Institute for Advanced Nursing (RIAN), Dong Nai Technology University, Bien Hoa 810000, Vietnam; 16Department of Health Policy and Management, College of Health Science, Korea University, Seoul 02841, Republic of Korea; 17Faculty of Public Health, VNU University of Medicine and Pharmacy, Vietnam National University, Hanoi 123080, Vietnam; 18International Institute for Training and Research (INSTAR), VNU University of Medicine and Pharmacy, Vietnam National University, Hanoi 100000, Vietnam; 19Center for Digital Health, Medical Science Research Institute, Kyung Hee University College of Medicine, Seoul 02448, Republic of Korea; 20Department of Pediatrics, Kyung Hee University Medical Center, Kyung Hee University College of Medicine, Seoul 02447, Republic of Korea

**Keywords:** mental health, depression, depressive disorders, mood, diet, nutrition, nutritional psychiatry, psychiatry

## Abstract

Background: Different patterns of food consumption may be associated with a differential risk of depression. Differences in dietary patterns between men and women and across different age groups have been reported, but their influence on the risk of depression has not been fully explored. Objectives: To investigate the associations between dietary patterns and risk of depression across sex and age groups to identify vulnerable subpopulations, which may inform targeted prevention and intervention strategies. Methods: The ALIMENTAL study was a cross-sectional, online international survey conducted between 2021 and 2023. Dietary data were collected using a validated food frequency questionnaire; depression data were collected using a self-reported validated questionnaire. Principal component analysis (PCA) was applied to identify distinct food consumption patterns. Multivariate analyses were then conducted to assess the associations between these patterns and depression, adjusting for multiple potential confounders. Results: Among 15,262 participants without chronic diseases or current psychotropic treatments, 4923 (32.2%) were classified in the depression group. Among those aged 18–34, the PCA-derived factor of ultra-processed foods consumption was significantly associated with increased risk of depression in both sexes with similar odds ratios (women 1.21, 95% confidence interval (CI): (1.15; 1.27), men 1.21, 95% CI: (1.07–1.18)). In women aged 18–34, the PCA factors for sodas (aOR 1.10, 95% CI: (1.06; 1.95) and canned and frozen foods (aOR 1.10, 95% CI: (1.04; 1.15) were associated with an increased risk of depression. In participants aged 35–54 years, the association between ultra-processed foods and depression was only observed in women (35–54 years: aOR 1.30, 95% CI: (1.20; 1.42), ≥55 years: 1.41, 95% CI: (1.11; 1.79)), with a significant association between a higher adherence to the PCA-derived “healthy diet” factor (e.g., fruits, nuts, green vegetables) and a lower risk of depression (35–54 years: aOR 0.82, 95% CI: (0.75; 0.89), ≥55 years: aOR 0.79, 95% CI: (0.64; 0.97)). Conclusions: These results show significant differences between men and women and between age groups regarding associations between dietary patterns and the risk of depression. These findings can help better target public health interventions.

## 1. Introduction

Dietary exposures have consistently been associated with differing risks of depressive symptoms and incident depression. Dietary patterns refer to the overall combination and quantities of foods and beverages habitually consumed, reflecting the complexity of the diet as a whole. A recent umbrella review of observational meta-analyses focused on dietary patterns associated with the risk of depression and concluded that the strongest level of evidence was found for an association between adherence to a Mediterranean dietary pattern—rich in fruits, vegetables, legumes and olive oil—and a lower risk of depression. Conversely, a Western diet characterized by ultra-processed foods and high levels of fat, sugar, and energy was associated with a higher risk, although with weaker evidence [1]. Mendelian-randomization meta-analyses also suggest that dietary patterns, and more specifically low daily fruit consumption [2] may influence the risk of depression [3]. Mediterranean diet interventions improved depressive symptoms in participants aged 22 to 53 years with moderate to severe depression, according to a meta-analysis of five randomized controlled trials involving 1507 participants [4].

Five recent meta-analyses concluded that the consumption of ultra-processed foods —formulations made mostly or entirely from substances derived from foods and additives, with little to no intact whole foods—was associated with poor physical or mental health outcomes [5,6,7,8,9]. However, some of these meta-analyses reported significant heterogeneity, which could be explained in part by differences between sexes and age groups.

Women have a higher risk of depression than men. This may in part be explained by certain dietary patterns being more common in women, including emotional eating, which involves consuming large quantities of high-calorie foods, particularly those rich in sugars [10,11,12,13,14]. Youth are particularly exposed to sweetened sodas and ultra-processed food consumption [15,16].

Over recent decades, the nature of consumed foods and the nutritional quality of basic ingredients have shifted due to the industrialization of the global food system and the increasing adoption of Western dietary patterns. This means that individuals probably have not been exposed to the same dietary factors, depending on whether they were 20 years old, for example, in 1980 or in the year 2000. Certain dietary patterns may produce progressive harm over the life course, emphasizing the need to define and study specific age groups to distinguish associations between dietary patterns and the risk of depression.

The objective of this study was to investigate the associations between dietary patterns and major depression across sex- and age-specific groups to identify potentially vulnerable subpopulations, which may inform targeted prevention and intervention strategies.

## 2. Materials and Methods

### 2.1. Study Design and Participants

The ALIMENTAL study was a cross-sectional, online international survey distributed among the adult population aged 18 and over between November 2021 and June 2023. An online self-questionnaire was made available via the FramaForm1^®^ platform after ethical committee approval (Est 1 N° SI 20.12.07.50145#1—IDRCB 2020-A03336-331). The study was disseminated via social media, mental health associations (UNAFAM network in France, Schizo-Oui!, Promesses, Argos2001), in hospitals, and universities through professional and student mailing lists.

### 2.2. Ethical and Regulatory Aspects

The study was carried out in accordance with the ethical principles for human medical research (World Medical Association Declaration of Helsinki), with the reference methodology MR003 of the CNIL and with approval of an approved ethical review board for France and Germany: Est 1 N° SI 20.12.07.50145#1—IDRCB 2020-A03336-331 and for Canada: CÉRUL Santé 2021-463.

Participation in the questionnaire was voluntary and anonymous. Participants could also withdraw from the study at any time. All respondents to the online questionnaire agreed to take part in the study. Participants were informed at the time of clicking that they were giving their consent for the data to be used for scientific publication purposes.

### 2.3. Inclusion and Exclusion Criteria

Participants were included if they were aged 18 or over. Participants were excluded if (i) they reported having at least one chronic illness among type 1 or 2 diabetes, hypertension, epilepsy, asthma, lung, colon, breast, or other cancer, chronic renal failure, chronic obstructive pulmonary disease, obstructive sleep apnea syndrome, irritable bowel syndrome, celiac disease, Crohn’s disease, hemorrhagic rectocolitis, or other chronic pathology with disability; or (ii) they reported being currently treated by antidepressants, mood stabilizers or antipsychotics. An exhaustive list of molecules from each of these classes with their International Nonproprietary Names and trade names (adapted to each country) was provided to the participants so they could determine if their current treatment included any of these three classes. The purpose of these two exclusion criteria was (i) to eliminate confounding factors that could have altered the associations between dietary patterns and depression; and (ii) to minimize the reverse causation effect of mood disorders on dietary patterns. For example, people with diabetes are at a higher risk of depression although they consume lower levels of glycemic sugars, and this could have biased the results [17]. Participants treated with antidepressants, mood stabilizers, and antipsychotics were also excluded to eliminate the influence of psychotropic drugs on the observed phenomena, and because the presence of these treatments is collinear with the presence of depression.

### 2.4. Collected Variables

#### 2.4.1. Depression Group

The risk of depression was assessed using the validated 20-item CES-D (Center for Epidemiologic Studies Depression Scale) [18]. Participants were categorized in the “depression” group if they scored ≥ 20. This cut-off offers the best compromise between sensitivity and specificity for defining a probable current major depressive episode [19].

#### 2.4.2. Dietary Pattern

The PNNS questionnaire (Programme National Nutrition Santé) [20] is a dietary assessment tool designed to evaluate adherence to French national nutritional guidelines. It consists of a series of questions assessing food consumption frequency. The questionnaire covers various food groups, including fruits and vegetables, dairy foods, whole grains, proteins, and added sugars. The questionnaire comprises 43 items divided into six categories: carbohydrates (8 items), proteins (9 items), fibers (8 items), omega-3 and saturated fats (7 items), ultra-processed foods (10 items), and drinks (7 items). This tool has been validated in epidemiological studies and is widely used in public health research to examine the relationship between diet and health outcomes. Items for ultra-processed food as defined using the Nova food classification system [21] were added to complete the questionnaire exploring the frequency (with a Likert-7 scale: less than once a week/1 to 2 times a week/3 to 4 times a week/5 to 6 times a week/once a day/2 to 3 times a day/4 times a day or more) of consumption of specific foods over the previous month. The questionnaire is presented in Appendix A.

#### 2.4.3. Sociodemographic, Health, and Lifestyle Variables

Referring to previous population-based studies focusing on dietary patterns and depression [22], the following additional sociodemographic, health, and lifestyle variables were reported:
−sex (binary);−body mass index (kg/m^2^) (calculated based on the self-reported height and weight, with obesity being defined by a body mass index ≥ 30);−presence of a partner living at home (binary);−presence of children at home (binary);−education level (university/tertiary vs. high school or lower degree) (binary);−unemployment (binary);−number of cigarettes smoked daily. Participants were classified in the “current daily smoking” group if they answered that they currently smoked one or more cigarette per day;−phototype using the Fitzpatrick classification. Phototypes were included because they can influence vitamin D status and sun exposure, which are potential confounding factors that may affect depressive status. Phototypes 1 and 2 (redheads and blondes) were grouped together and compared to the other 4 combined phototypes;−subjective nutritional knowledge. Participants answered the question “Do you consider yourself to have good knowledge of nutrition?” They were classified in the “good knowledge” group if they answered “yes” or “rather yes,” and in the other group if they answered “no” or “rather no.”;−physical activity. Participants were classified in the “physically inactive” group according to the Moderate to Vigorous Physical Activity (MVPA) score on the validated “Observatoire National de l’Activité Physique et de la Sédentarité” (National Observatory of Physical Activity and Sedentary Behavior) Physical Activity Questionnaire (ONAPS-PAQ). The questionnaire measures, based on self-reported values, the volume of physical activity and sedentary time of the respondent during a typical week.

### 2.5. Statistical Analysis

A descriptive analysis was carried out to describe the main characteristics of the overall sample. Categorical variables were presented as numbers and percentages, and continuous variables as means ± standard deviation (SD). In order to identify dietary patterns, we performed a principal component analysis (PCA) with varimax rotation to reduce the large number of food frequency items to a smaller number of uncorrelated food intake components. The number of components was chosen based on the Kaiser stopping criterion (i.e., all components with eigenvalues greater than 1) and the screen test. The use of component scores as the independent variables in multivariable models is considered as relevant in the case of multicollinearity [23]. The dietary factors were thus used for multiple regression analysis, associated to a set of confounding factors selected from the univariable analysis, with selection based on a threshold *p*-value < 0.20 or if clinically relevant. The following variables were included as confounding variables: obesity, current daily smoking, education level, nutrition knowledge, partner living at home, children at home, phototype, employment status, and being physically inactive. Six multiple logistic regression models were then built for 6 subgroups determined by sex and age ([18–34], [35–54] and [55+] years). These three age groups were determined to homogenize dietary patterns. We then applied a Benjamini–Hochberg correction in the multivariate models due to the high number of regressions and analyses. Odds ratios (ORs) and 95% confidence intervals (95%CI) were reported. The 13 dietary factors were selected based on eigenvalues greater than 1 (Kaiser criterion) and inspection of the scree plot; food items with absolute loadings ≥ 0.3 were grouped, and factors were named according to the predominant foods with the highest loadings. Specifically, two authors (Yannis Achour and Guillaume Fond) independently reviewed the rotated component loadings and identified clusters of food items that shared common nutritional or processing characteristics. They then assigned each factor a descriptive label—such as “Ultra-Processed Foods” or “Healthy Diet”—based on these shared features and guided by terminology used in previous PCA-based dietary pattern publications. Any discrepancies in naming were resolved through discussion until full consensus was reached.

A two-sided *p*-value of less than 0.05 was considered to indicate statistical significance. The statistical analyses were performed using R software (version 4.3.3) and SPSS Statistics for Windows (Version 21.0. IBM Corp, Armonk, NY, USA).

## 3. Results

A total of 15,262 participants were included in the study. The distribution of participants by country is presented in Appendix A. Overall, 13,107 (85.9%) participants were women, the mean age was 33 ± 12.8 years, 4923 (32.2%) were classified in the depression group, 1514 (9.9%) were classified in the “obesity” group, 2690 (17.6%) in the current daily smoking group, 2149 (14.1%) were physically inactive, 8672 (56.8%) had a university/tertiary level, and 11,971 (78.4%) reported good nutrition knowledge.

The sample’s socio-demographic characteristics and the six subgroups (by sex and age category) are presented in Table 1 and Table 2, respectively. Table 3 shows the 13 dietary pattern factors identified in the principal component analysis, including healthy diet (Factor 1), ultra-processed foods (Factor 2), starchy foods (Factor 3), alcohol and coffee (Factor 4), eggs (Factor 5), meat (Factor 6), high glycemic index foods and processed fat (Factor 7), supplements (PUFAs and proteins), chia seeds and oat flakes (Factor 8), dairy foods and fruit juice (Factor 9), canned and frozen foods (Factor 10), fish (Factor 11), sugary or sweetened sodas (Factor 12), and decaffeinated coffee (Factor 13).

In the multivariable models and after post-multiple testing (Table 4, univariable analyses in Appendix A), Factor 1, labeled “healthy diet,” was associated with a lower risk of depression in women aged 18–34 years (adjusted odds ratio (aOR) 0.84, 95% CI [0.80; 0.89], *p* < 0.001) and 35–54 years (aOR 0.82, 95% CI [0.75–0.89], *p* < 0.001). Healthy diet included fruits, nuts (almonds or hazelnuts), green vegetables (e.g., green beans, broccoli, asparagus), olive oil, rapeseed, or soybean oil, leafy salad or endives, tea consumption, whole-grain bread, and cheese. Factor 2, “ultra-processed foods”, including chips, salty biscuits, fried foods, pastries, cakes, sweet biscuits, junk food, pre-cooked meals, industrially processed meat, and roasted snack seeds, was significantly associated with an increased risk of depression in both sexes aged 18–34 years (women aOR 1.21, 95% CI [1.15; 1.27], *p* < 0.001; men aOR 1.21, 95% CI [1.07; 1.18], *p* < 0.027).

These associations were also observed in older women, with higher odds ratios (women aged 35–54 years aOR 1.30, 95% CI [1.20; 1.42], *p* < 0.001, over 55 years aOR 1.41, 95% CI [1.11; 1.79], *p* < 0.02).

Other dietary factors were significantly associated with a higher risk of depression, especially in women; these included canned and frozen foods (Factor 10) (aOR 1.10, 95% CI [1.04; 1.15], *p* < 0.002), sugary or sweetened sodas (Factor 12) (aOR 1.07, 95% CI [1.03; 1.13], *p* < 0.012), and dietary supplements (omega-3 fatty acids and proteins), chia seeds, and oat flakes (Factor 8) (aOR 1.14, 95% CI [1.06; 1.23], *p* < 0.001).

In women aged 18–34 years, the consumption of high-glycemic-index foods and processed fats (Factor 7) (aOR 0.92, 95% CI [0.87; 0.96], *p* < 0.005), dairy foods and fruit juices (Factor 9) (aOR 0.92, 95% CI [0.87; 0.96], *p* < 0.005), and fish (Factor 11) (aOR 0.93, 95% CI [0.88; 0.98], *p* < 0.025) were significantly associated with lower risk of depression.

Some confounding factors were associated with a reduced risk of depression in certain groups (higher attainment, nutrition knowledge, having a partner living at home, and children at home), while others were associated with an increased risk of depression (current daily smoking, obesity, unemployment, and phototypes 1 and 2).

## 4. Discussion

Our results highlighted that specific dietary patterns were found to be associated with risk of depression in this large study. Our study reported that for women, the ultra-processed food factor (Factor 2) increased the odds of depression whatever the age group considered, while in men this association was only observed in the 18–34 y age group. We also reported that the reduced odds ratio of depression associated with healthy diet factor was observed in women aged 18–54 years but not in men. Our findings suggest that the associations between dietary patterns and depression risk vary by sex and age group. In line with recent guidelines [24], these results highlight the importance of identifying when, and in whom, specific dietary patterns may offer protection against depression, to inform the design of tailored nutritional interventions.

The association between higher consumption of ultra-processed foods and higher risk of depression in all age groups in women reinforces the findings of two recent meta-analyses published in 2021 and 2022, confirming the association between respectively higher consumption of ultra-processed foods [5,6,7,8,9] and sugar-sweetened sodas [25] and higher risk of depression. Several hypotheses could be formulated regarding the observed difference in the association between ultra-processed food consumption and depression among women and men. Women have different hormonal and metabolic profiles compared to men, which could make them more susceptible to poor health outcomes associated with ultra-processed foods on mental health [26]. Women and men may have different dietary habits in terms of types and quantities of ultra-processed foods consumed, and within genders there may be wider differences. These differences could mediate the divergence in the association with depression [27]. Ultra-processed foods may also alter the gut microbiota, which is connected to the brain through the gut-brain axis [28]. Differential microbiota composition between women and men could contribute to distinct levels of susceptibility to depression related to these foods [29]. Women and men may respond differently to messages about diet and health. These differences could influence perceptions of the consumption of ultra-processed foods and might have distinct sex-based impacts on mental health [30]. These differences could also explain why the association of soda and canned/frozen food factors with depression was reported to be significant only in women, not in men. In addition, by selecting participants without chronic diseases, amongst which some are more common in women, it is less likely that chronic diseases might explain the sex difference of the association we reported. Despite our efforts to standardize the group distribution, our sample is unbalanced in favor of women. This is a normative finding in the depression literature and may be influenced by the fact that our study was distributed in many healthcare facilities (where staff were predominantly female) as well as on social media.

Different age groups might have distinct dietary patterns and a differential exposure to ultra-processed foods. A recent systematic review concluded that higher ultra-processed food intakes were associated with younger age, urbanization and being unmarried, single, separated or divorced [31]. Younger generations are likely to have been exposed to these foods from an earlier age, potentially supporting the stronger association in younger men than older ones [32]. Importantly, higher consumption of ultra-processed food has been associated with multiple chronic conditions that are often comorbid with depression and relate particularly to inflammation, metabolic, and cardiovascular health [33], suggesting the importance of disseminating information against their consumption notably in individuals with depression [34].

The “healthy diet” factor (Factor 1) shared several characteristics with the Mediterranean diet in terms of composition. The latest meta-analysis of the effectiveness of the Mediterranean diet interventions in depression suggested significant heterogeneity and the need for high-quality trials in large samples [4]. Our results suggest that women could be prioritized as a target and that stratifying effectiveness by age range might be useful.

Regarding the association between higher consumption of oat flakes, omega-3 supplements, protein supplements, and chia seeds (Factor 8) and higher risk of depression, it is likely that this reflects an indication bias and reverse causality—meaning that individuals with depression may be more inclined to incorporate these foods and supplements into their diet in an effort to alleviate their depressive symptoms. It cannot be ruled out that the consumption of these foods is inversely associated with the exclusion of other foods, which could also help explain the observed association.

These results suggest that guidelines for the treatment of depression could be informed by age and sex. Greater nutritional knowledge in women aged 18 to 54 years who participated in our study was associated with a lower risk of depression. This finding may encourage the promotion of nutrition knowledge through targeted prevention programs for women with depression or at risk of depression. More research is needed, especially in larger samples with adequate representation of both men and women.

The results of this study need to be viewed within the context of the following limitations: First, to address causality limitations inherent in a cross-sectional design, we emphasize the need for future prospective cohort studies and randomized intervention trials to confirm these associations and reduce recall bias. Due to the cross-sectional design, we cannot infer causality; there remains the possibility of reverse causation, whereby depressive symptoms influence dietary behaviors, underlining the need for longitudinal and interventional studies. Second, as is the case with many national surveys, we cannot calculate the participation rate because we do not know the number of individuals who received (and opened) the link for the questionnaire. Given the sample’s skewness toward women (85.9%), the generalizability of findings to men is limited; future research should aim for more balanced sex representation. The limited representation of participants aged ≥ 55 years (7% of the sample) might affect the robustness of findings in this age group and should be considered in interpretation. As the questionnaire was distributed in French, this limited participation in non-Francophone countries. However, our objective was not to describe the dietary patterns of the population, but to explore the associations with the risk of depression. Third, we used a semi-quantitative food questionnaire that covered only specific food items and is acknowledged to be less precise than dietary assessment through food diaries. Moreover, no nutrient estimation was performed, thus precluding any evaluation of total energy intake, which could represent a potential confounding factor in the assessment of dietary patterns and depression. However, to our knowledge, no studies investigating the association between dietary patterns and depression risk have reported a major influence of total energy intake. Fourth, depression was assessed using a self-reported scale, which may not be directly comparable to a clinical diagnosis. Nevertheless, the CES-D scale has been shown to be a reliable and valid instrument for measuring depressive symptoms. Additionally, inclusion of nutrient-level analyses, such as total energy intake, is recommended to better adjust for dietary confounders. Reliance on self-reported dietary and depression measures may introduce recall and social desirability biases; future research should incorporate objective biomarkers and detailed dietary records, including energy and nutrient quantification, to elucidate underlying mechanisms. Without biomarker validation or prospective follow-up, mechanistic interpretations remain speculative; future cohort studies and mechanistic trials are required to confirm these associations. We acknowledge that BMI does not distinguish lean from fat mass or reflect fat distribution; future studies should include measures such as waist-to-hip ratio or dual-energy X-ray absorptiometry.

## 5. Conclusions

In a population free of chronic diseases affecting dietary patterns and not in receipt of psychotropic treatment, associations between dietary patterns and depression risk were observed, with significant differences being identified with respect to sex and age groups. Higher ultra-processed food consumption was associated with increased depression risk in individuals of both sexes aged 18–34 years, and in women aged 35 years and older. Higher adherence to a healthy dietary pattern was associated with lower depression risk only in women aged 18–54 years. Although cross-sectional, these findings provide further justification for interventions in specific subpopulations to optimize the prevention of depression.

## Figures and Tables

**Table 1 nutrients-17-01583-t001:** Sample characteristics.

Sociodemographic Characteristics	N = 15,262	%
Sex (woman)	13,107	85.9
Age (Years, mean ± SD)	33 ± 12.8	
Age min–max value	18–92	
Obesity	1514	9.9%
Current daily smoking	2690	17.6%
Education level (academic level)	8672	56.8%
Nutrition knowledge	11,971	78.4%
Partner living at home	7634	50.0%
Children at home	5395	35.3%
Unemployment	2368	15.5%
Phototype 1 and 2 vs. other phototypes	3912	25.6%
Physically inactive	2149	14.1%

**Table 2 nutrients-17-01583-t002:** Breakdown of the six groups by sex and age category.

Age Category	Number of Persons (% of the Whole Sample)	Men (% of the Age Group)	Women (% of the Age Range)
18–34 years	9099 (60%)	1187 (13%)	7912 (87%)
35–54 years	5049 (33%)	776 (15%)	4273 (85%)
≥55 years	1114 (7%)	192 (17%)	922 (83%)

**Table 3 nutrients-17-01583-t003:** Dietary patterns factors derived in the principal component analysis.

Factor 1 Healthy diet	Fruit
	Nuts, almonds, or hazelnuts (handful of about six units)
	Green vegetables (green beans, broccoli, asparagus, etc.)
	Tablespoon of olive oil, rapeseed oil, or soybean oil
	Green salad or endives
	Tea (cup)
	Whole-grain bread
	Cheese
Factor 2 Ultra-processed foods	Chips, savory biscuits
	Fried foods (frozen or non-frozen fries, fish sticks, nuggets, cordon bleu)
	Pastries, cakes, French baked goods, sweet biscuits
	“Junk food” (McDonald’s, KFC, Burger King, Quick, sandwich, kebab, quiches, etc.)
	Pre-cooked meals (canned, packaged, frozen meals)
	Industrially processed meat (salting, maturation, fermentation, smoking): ham, slices of chicken breast, turkey, hotdogs (Frankfurt sausages), corned beef, dried beef, canned meats, sausages
	Roasted snack seeds (peanuts, almonds, roasted pistachios)
Factor 3 Starchy foods	Whole or semi-whole rice
	Whole or semi-whole pasta
	Quinoa, bulgur, semolina
	Legumes (lentils, chickpeas, white beans, red beans)
Factor 4 Alcohol and coffee	Beer (half-pint or 25cl)
	Red wine (glass)
	Other wine (glass)
	Spirits (1 glass or 3cl)
	Coffee (cup)
Factor 5 Eggs	Egg yolk
	Egg white
Factor 6 Meat	Unprocessed white meat (chicken, turkey, rabbit)
	Red meat (beef, veal, pork, lamb, mutton, horse, goat, duck)
Factor 7 High-glycemic-index foods and processed fat	Potato (baked, boiled), white pasta, fresh pasta (= non-whole and non-semi-whole), white rice
	Jam, spread (teaspoon)
	White bread, cereal toast (excluding oat flakes)
	Butter (including cooking)
	Margarine (including cooking)
Factor 8 Supplements (PUFAs and proteins), chia seeds and oat flakes	Protein dietary supplements (powders, tablets, protein bars, or other forms)
	Chia seeds
	Omega-3 (in capsules or syrup)
	Oat flakes
Factor 9 Dairy foods and fruit juice	Fruit yogurt or flavored yogurt or fromage blanc with fruits or flavored
	Plain yogurt or plain fromage blanc
	Dessert cream, mousse, sweet dessert, ice cream
	Glass of milk (whole, skimmed, or semi-skimmed)
	Fruit juice (pressed, industrial) excluding lemon juice
Factor 10 Canned and frozen foods	Uncooked canned foods (e.g., corn, white beans) excluding meat and fish
	Uncooked frozen foods (e.g., peas, green beans) excluding meat and fish
Factor 11 Fish (fatty and lean)	Other fish and seafood (not listed above) including canned and frozen
	Raw fatty fish (sardines, salmon, mackerel, tuna (only fresh or canned, not frozen))
Factor 12 Sugary or sweetened sodas	Diet or zero-calorie soda
	Sugary sodas
Factor 13 Decaffeinated coffee	Decaffeinated coffee (cup)

**Table 4 nutrients-17-01583-t004:** Multivariable analyses of the associations between 13 dietary patterns and depression in 15,262 participants without chronic physical illness and without psychotropic treatment. The *p*-value was corrected using the Benjamini–Hochberg method. F—Factor calculated in the Principal Component Analysis. Depression was defined by a Center for Epidemiologic Studies Depression Scale (CES-D) score ≥ 20. Significant associations between dietary patterns and an increased risk of depression (OR > 1, *p* < 0.05) are highlighted in red; significant associations between dietary patterns and a decreased risk of depression (OR < 1, *p* < 0.05) are highlighted in green.

	Whole Sample (n = 15,262)																							
	Women (n = 13,107; 85.9%)												Men (n = 2155; 14.1%)											
	18–34 y				35–54 y				≥55 y				18–34 y				35–54 y				≥55 y			
	n = 7912				n = 4273				n = 922				n = 1187				n = 776				n = 192			
	n (CES D ≥ 20) = 3201 (40%)				n (CES D ≥ 20) = 1050 (25%)				n (CES D ≥ 20) = 179 (19%)				n (CES D ≥ 20) = 321 (27%)				n (CES D ≥ 20) = 151 (19%)				n (CES D ≥ 20) = 21 (11%)			
	OR	(95% IC)		*p*	OR	(95% IC)		*p*	OR	(95% IC)		*p*	OR	(95% IC)		*p*	OR	(95% IC)		*p*	OR	(95% IC)		*p*
Dietary pattern factors derived from principal component analysis																								
**F1 Healthy diet**	**0.84**	**0.80;**	**0.89**	**<0.001**	0.82	0.75;	0.89	**<0.001**	0.79	0.64	0.97	0.084	0.91	0.76;	1.08	0.474	1.18	0.93;	1.48	0.346	0.77	0.38;	1.52	0.675
F2 Ultra-processed foods	**1.21**	**1.15;**	**1.27**	**<0.001**	**1.30**	**1.20;**	**1.42**	**<0.001**	**1.41**	**1.11**	**1.79**	**0.027**	**1.21**	**1.07;**	**1.18**	**0.018**	0.93	0.74;	1.15	0.692	1.19	0.45;	3.03	0.825
F3 Starchy foods	1.02	0.97;	1.07	0.610	1.11	1.02;	1.20	0.053	1.07	0.87	1.30	0.695	1.02	0.89;	1.18	0.833	0.89	0.72;	1.09	0.474	0.39	0.14;	0.92	0.093
F4 Alcohol and coffee	0.93	0.88;	0.99	0.053	1.09	1.01;	1.18	0.098	1.13	0.95	1.35	0.353	0.94	0.83;	1.06	0.533	1.10	0.95;	1.27	0.385	1.10	0.72;	1.66	0.778
F5 Eggs	1.01	0.96;	1.06	0.859	**1.11**	**1.03;**	**1.20**	**0.029**	1.04	0.85	1.26	0.815	1.06	0.93;	1.20	0.603	1.01	0.83;	1.21	0.947	0.63	0.29;	1.21	0.358
F6 Meat	1.01	0.96;	1.05	0.890	0.95	0.87;	1.03	0.406	0.94	0.76	1.16	0.726	1.03	0.90;	1.17	0.815	1.19	0.98;	1.45	0.197	0.74	0.29;	1.67	0.683
F7 High-glycemic-index foods and processed fat	0.92	0.87;	0.96	**0.005**	1.01	0.94;	1.09	0.815	1.06	0.90	1.24	0.683	0.85	0.73;	0.99	0.105	1.00	0.81;	1.21	0.959	0.84	0.45;	1.48	0.718
F8 Supplements (PUFAs and proteins), chia seeds and oat flakes	1.06	1.01;	1.12	0.074	1.14	1.06;	1.23	**0.004**	1.09	0.93	1.27	0.517	0.96	0.84;	1.09	0.720	0.99	0.81;	1.18	0.931	1.34	0.74;	2.36	0.540
F9 Dairy foods and fruit juice	0.92	0.87;	0.96	**0.003**	0.93	0.87;	1.01	0.197	0.95	0.79	1.14	0.754	0.89	0.77;	1.02	0.223	1.03	0.83;	1.26	0.867	0.82	0.45;	1.41	0.683
F10 Canned and frozen foods	1.10	1.04;	1.15	**0.002**	1.02	0.95;	1.10	0.726	1.05	0.87	1.27	0.726	1.17	1.02;	1.34	0.092	1.07	0.87;	1.32	0.692	0.67	0.28;	1.46	0.540
F11 Fish (fatty and lean)	0.93	0.88;	0.98	**0.025**	0.91	0.85;	0.99	0.070	1.17	0.99	1.39	0.165	0.88	0.76;	1.02	0.208	1.08	0.89;	1.31	0.647	2.17	1.15;	4.46	0.059
F12 Sugary or sweetened sodas	1.07	1.03;	1.13	**0.012**	1.04	0.96;	1.13	0.540	0.94	0.75	1.17	0.726	1.14	1.00;	1.30	0.152	0.92	0.74;	1.14	0.654	1.56	0.56;	3.91	0.597
F13 Decaffeinated coffee	1.05	0.99;	1.11	0.249	1.01	0.95;	1.09	0.815	0.92	0.79	1.06	0.461	0.99	0.83;	1.17	0.935	1.13	0.94;	1.36	0.366	0.91	0.39;	1.77	0.867
**Confounding factors**																								
Obesity	1.38	1.16;	1.64	**0.002**	1.32	1.07;	1.62	**0.037**	1.02	0.55	1.83	0.947	1.54	0.82;	2.83	0.361	1.25	0.66;	2.28	0.683	0.87	0.08;	7.69	0.935
Current daily smoking	1.48	1.30;	1.69	**<0.001**	1.23	1.02;	1.49	0.097	1.42	0.87	2.28	0.342	1.44	0.99;	2.09	0.150	0.81	0.49;	1.32	0.619	0.47	0.02;	3.50	0.685
Education level (academic level)	0.79	0.71;	0.88	**<0.001**	0.87	0.72;	1.04	0.265	1.12	0.71	1.83	0.778	0.94	0.69;	1.28	0.815	0.52	0.33;	0.84	**0.035**	0.92	0.18;	6.08	0.947
Nutrition knowledge	0.80	0.72;	0.90	**0.002**	0.69	0.55;	0.87	**0.012**	0.91	0.46	1.95	0.867	0.69	0.48;	0.99	0.114	0.94	0.54;	1.71	0.885	0.76	0.13;	4.92	0.855
Partner living at home	0.66	0.59;	0.74	**<0.001**	0.77	0.65;	0.92	**0.022**	0.68	0.47	0.99	0.121	0.49	0.34;	0.70	**0.001**	0.54	0.33;	0.87	**0.045**	1.37	0.30;	8.35	0.815
Children at home	0.64	0.54;	0.76	**<0.001**	0.90	0.74;	1.10	0.529	1.33	0.90;	1.95	0.323	0.75	0.39;	1.37	0.578	0.74	0.47;	1.18	0.385	0.83	0.19;	3.18	0.867
Unemployment	1.25	1.11;	1.41	**0.002**	1.46	1.17;	1.81	**0.006**	2.45	1.23	4.76	**0.047**	1.95	1.35;	2.79	**0.003**	1.67	0.94;	2.89	0.193	6.68	0.91;	45.6	0.154
Phototype 1 and 2 vs. other phototypes	1.25	1.12;	1.39	**0.001**	1.11	0.93;	1.31	0.449	1.16	0.75	1.76	0.685	1.47	1.04;	2.06	0.088	2.08	1.32;	3.24	**0.011**	5.93	1.24;	28.5	0.086
Physically inactive	1.02	0.89;	1.18	0.855	1.42	1.18;	1.70	**0.002**	1.84	1.10	3.03	0.070	1.19	0.73;	1.93	0.683	2.19	1.27;	3.70	**0.025**	2.20	0.35;	12.0	0.603

## Data Availability

Data are available on reasonable request.

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
