# Peer review of "Dietary Patterns and Major Depression: Results from 15,262 Participants (International ALIMENTAL Study)"

_nutrients, 2025, doi:10.3390/nu17091583_

Round 1

Reviewer 1 Report

Comments and Suggestions for Authors

The manuscript titled "Dietary patterns and major depression. Results from 15,262 participants (international ALIMENTAL study)" presents a comprehensive cross-sectional analysis of the associations between dietary patterns and depression risk across sex and age groups. While the study addresses an important and timely topic, several methodological and presentation issues need to be addressed before the manuscript can be considered for publication. Below are my detailed comments and suggestions for improvement.

Major Concerns:

The study design is cross-sectional, which inherently limits the ability to infer causality between dietary patterns and depression risk. The authors acknowledge this limitation (Page 13), but the discussion could further emphasize the need for prospective or intervention studies to confirm these associations. Additionally, the reliance on self-reported dietary data (Page 4) introduces potential recall bias, and the lack of nutrient-level analysis (e.g., total energy intake) is a notable omission. The authors should discuss how these limitations might affect the interpretation of the results.

The sample is heavily skewed toward women (85.9%, Page 6), which raises concerns about the generalizability of the findings to men. While the authors note that this is common in depression research (Page 12), the implications of this imbalance for the study's conclusions should be more thoroughly discussed. Furthermore, the representation of older adults (≥55 years) is limited (7% of the sample, Page 6), which may affect the robustness of the findings for this age group.

Page 4, Sociodemographic, health and lifestyle variables, there should be reason for the authors to state why they choose theses foctors as sociodemographic factors, so maybe it would be better for the author to refer to some relevant references to support their selection of factors. For example, the author can revise the statement as: “Sociodemographic, health and lifestyle variables. Referring to previous population-based studies focusing on dietary patterns and depression [doi: 10.1139/apnm-2023-0550], the following additional variables were reported.”

Moreover, The manuscript would benefit from a clearer explanation of the Principal Component Analysis (PCA) methodology. For instance, the criteria for selecting the 13 dietary factors (Page 6) are not fully elaborated, and the rationale for grouping certain foods together (e.g., Factor 8: supplements, chia seeds, and oat flakes) is unclear. The authors should provide more details on the PCA process, including how the factors were named and interpreted, to enhance transparency and reproducibility.

Minor Concerns:

The tables and supplementary materials are extensive but could be more reader-friendly. For example, Table 4 (Page 8) is dense and difficult to interpret due to the large number of factors and subgroups. Simplifying the presentation or breaking it into smaller tables focused on key findings would improve clarity. Additionally, the supplementary questionnaire (Page 17) could be formatted more clearly to distinguish between response options and food items.

The discussion section (Pages 11–13) could be more focused. While the authors provide plausible explanations for the observed sex and age differences, some hypotheses (e.g., hormonal and metabolic profiles, Page 12) are speculative and lack direct support from the data. The discussion would benefit from a tighter connection to the study's findings and a more balanced consideration of alternative explanations.

Comments on the Quality of English Language

can be improved

Reviewer 2 Report

Comments and Suggestions for Authors

I am positive about research in this front. One of the main strengths of this study is its large sample size,  over 15,000 participants from multiple countries. This is a great positive point regarding the external validity of the findings and allows for meaningful subgroup analyses by age and sex. The exclusion of individuals with chronic diseases or treatment is another strength that minimizes confounding.

Methodologically, the study employs a robust approach by using Principal Component Analysis (PCA). This is an appropriate and well-established technique in nutritional epidemiology, allowing researchers to reduce complex food intake data into interpretable factors. Furthermore, the authors use multivariate logistic regression models, appropriately adjusted for key sociodemographic and lifestyle variables.

The findings are clearly reported, with adjusted odds ratios and confidence intervals provided for each subgroup.

But, despite its strengths, the manuscript has several limitations that I would like to discussion with the authors. I hope might comments are of interest the improve the current paper.

First, its cross-sectional is informative, and the direction of the relationship between diet and depression cannot be confirmed. Individuals with depression might adopt certain dietary patterns as a coping mechanism, rather than poor diet being the cause of depression. This should be carefully discussed in the manuscript.

Second, the study relies on self-reported data for both dietary intake and depression symptoms. Although validated questionnaires were used (CES-D for depression and PNNS for diet), self-reporting is related to bias and social desirability bias. The dietary assessment does not include detailed nutrient intake or caloric data, limiting the analysis and the ability to assess mechanisms related to nutrient deficiencies or excesses.

The sample is also biases toward women (over 85%). This sex difference could limit the generalizability of findings to men and may have influenced the strength of observed sex differences. Moreover, the study does not report participation rates, and the recruitment method  could have introduced selection bias, favoring individuals with a higher interest in nutrition or mental health.

Furthermore no biomarker data or longitudinal follow-up is available to test these hypotheses. This makes the mechanistic interpretations more speculative than definitive.

Minor:

Another limitation worth noting is the reliance on Body Mass Index (BMI) as a measure of body composition. While BMI is widely used in epidemiological studies due to its simplicity, it is an imperfect for health and fat distribution. It does not distinguish between muscle and fat mass, nor does it account for fat localization. This should be duscussed

Additionally, Table 4 may be difficult for readers to interpret due to its dense format and the simultaneous presentation of multiple subgroups, dietary factors, and covariates. Although the use of adjusted odds ratios across age and sex groups is valuable, please, reconsider to provide the most relevant findings graphically (e.g., forest plots) or breaking the table into smaller, more digestible parts...

Round 2

Reviewer 1 Report

Comments and Suggestions for Authors

Good revisions.

Reviewer 2 Report

Comments and Suggestions for Authors

Thank you for addressing my comments